Two new genera and four new species of Asterocheridae (Copepoda: Siphonostomatoida) associated with sponges (Porifera) from the Korean East Sea

Kim Il-Hoi 1
Lee Taekjun leetj@syu.ac.kr 2 3
1 Korea Institute of Coastal Ecology , Bucheon , South Korea
2 Department of Animal Resources Science, Sahmyook University , Nowon-gu , Seoul , South Korea
3 Marine Biological Resource Institute, Sahmyook University , Nowon-gu , Seoul , South Korea
Semprucci Federica
Electronic publication date: 2023 Feb 21
Publication date: 2023
Volume: 11
Electronic Location ID: e14889
Received 2022 Nov 1; Accepted 2023 Jan 22
Copyright: ©2023 Kim and Lee
Copyright year: 2023
Copyright holder: Kim and Lee
License: This is an open access article distributed under the terms of the Creative Commons Attribution License, which permits unrestricted use, distribution, reproduction and adaptation in any medium and for any purpose provided that it is properly attributed. For attribution, the original author(s), title, publication source (PeerJ) and either DOI or URL of the article must be cited.
License URL: https://creativecommons.org/licenses/by/4.0/

Keywords: Amalomyzon n. gen., Dokdocheres n. gen., Asterocheres, Scottocheres, Taxonomy

Funding: National Institute of Biological Resources (NIBR) NIBR202227202 Ministry of Environment (MOE) Basic Science Research Program through the National Research Foundation of Korea (NRF) Ministry of Education 2021R1I1A2058017 This study was supported by a grant (NIBR202227202) from the National Institute of Biological Resources (NIBR) funded by the Ministry of Environment (MOE) and the Basic Science Research Program through the National Research Foundation of Korea (NRF) funded by the Ministry of Education (No. 2021R1I1A2058017), Republic of Korea. The funders had no role in study design, data collection and analysis, decision to publish, or preparation of the manuscript.

==============================
Two new genera and four new species of siphonostomatoid copepods of the family Asterocheridae associated with sponges are described from the Korean East Sea (Sea of Japan). These new copepods are distinguishable from related genera and species by their diagnostic morphological characters as follows: Amalomyzon elongatum n. gen. n. sp. bears an elongated body, two-segmented rami of legs 2, a uniramous leg 3 with two-segmented exopod, and a rudimentary leg 4 represented by a lobe. Dokdocheres rotundus n. gen. n. sp. has an 18-segmented female antennule, a two-segmented endopod of antenna, and unusual setations of swimming legs, including three spines plus four setae on the third exopodal segment of legs 2–4. Asterocheres banderaae n. sp. has no inner coxal seta on leg 1 or 4, but has two strong, sexually dimorphic inner spines on the second endopodal segment of male leg 3. Scottocheres nesobius n. sp. bears elongate female caudal rami about six times longer than wide, a 17-segmented female antennule, and two spines plus four setae on the third exopodal segment of leg 1.

Introduction

Siphonostomatoid copepods of the family Asterocheridae live in association with marine invertebrates. The most commonly reported hosts of these copepods are sponges, cnidarians, and echinoderms, although hosts are unknown for many species of copepods (Boxshall & Halsey, 2004). The great majority of the copepods of this family have been recorded from European and tropical seas with very few recorded in the entire Pacific region. Ho (1984) was the first one who recorded a species of the Asterocheridae in the Korean East Sea (Sea of Japan), who described Asterocheres aesthetes as a new species associated with sponge Spirastrella insignis Thiele, 1898 in Sado Island, Japan. Since then, seven additional species have been reported from the same sea (three associated with sponges, two with sea stars, and two with unknown hosts) (Table 1).

Table 1 Species of the family Asterocheridae recorded previously in the Korean East Sea.

Species	Hosts	Sources	
Asterocheres aesthetes Ho, 1984	The sponges Spirustrella insignis Thiele, 1898 and Suberites ficus (Johnston, 1842)	Ho (1984) and Kim (1998)	
Asterocheres cuspis Kim, 2016	A sponge of Myxila sp.	Kim (2016)	
Asterocheres lilljeborgi Boeck, 1859	The sea star Henricia leviuscula (Stimpson, 1857)	Kim (2016)	
Asterocheres simulans (Scott, 1898)	An unidentified sponge	Kim (2016)	
Asteropontoids acutirostris Kim, 2016	Host unknwon	Kim (2016)	
Callomyzon macrocephalum Kim, 2016	A Sponge epibiotic on the scallop Azumapecten farreri (Jones & Preston, 1904)	Kim (2016)	
Dermatomyzon nigripes (Brady & Robertson, 1880)	Host unknown	Kim (1998)	
Scottomyzon gibberum (Scott & Scott, 1894)	Various species of sea stars	Kim (1992)	

In the present article, two new genera and four new species of the Asterocheridae are described as associates of sponges in the Korean East Sea.

Materials & Methods

The copepods studied in the present work were extracted from sponges collected at Ulleung Island and Dokdo Island, the most remote islands of Korea located in the Korean East Sea (Sea of Japan) (Fig. 1). Sponge hosts were collected by trimix and SCUBA diving at depths of 22 m to 45.2 m. Copepods extracted from these sponge hosts were preserved in 80% ethanol. Permits for marine organisms collection and export were given by the county office of Ulleung-gun, Korea. For microscopic observation, copepods were immersed in lactic acid for at least 10 min and dissected. Dissected appendages were observed using the reverse slide method of Humes & Gooding (1964). Drawings were made under a light microscope equipped with a drawing tube. In the armature formula for the description of species, Roman numerals indicate spines and Arabic numerals represent setae. Lengths of copepod specimens and measurements of appendages were mostly based on a dissected and figured specimen of each species. Morphological terminology followed Huys & Boxshall (1991). Type specimens have been deposited in the National Institute of Biological Resources (NIBR), Incheon, Korea.

Figure 1 Map showing collection localities.

(A) Ulleung Island. (B) Dokdo Island.

This published work and nomenclatural acts it contains have been registered in ZooBank, an online registration system for the ICZN. The ZooBank LSIDs (Life Science Identifiers) can be resolved and the associated information can be viewed through any standard web browser by appending the LSID to the prefix http://zoobank.org/. The LSID for this publication is: [urn:lsid:zoobank.org:pub:0E84F5CE-D4C5-4AAD-A72C-CCBD77806CDB]. The online version of this work is archived and available from the following digital repositories: PeerJ, PubMed Central SCIE and CLOCKSS.

Results & Discussion

Taxonomy

Phylum Arthropoda von Siebold, 1848	
Subphylum Crustacea Brünnich, 1772	
Superclass Multicrustacea Regier, Shultz, Zwick, Hussey, Ball, Wetzer, Martin & Cunningham, 2010	
Class Copepoda Milne Edwards, 1840	
Order Siphonostomatoida Burmeister, 1835	
Family Asterocheridae Giesbrecht, 1899	
Amalomyzon n. gen.	
urn:lsid:zoobank.org:act:214F5279-8986-4F6A-A32E-FD3BCF0EA638	

Diagnosis. Body elongate, cylindrical, incompletely segmented. Urosome four-segmented; caudal ramus with six setae. Female antennule 20-segmented, with large aesthetasc on antepenultimate segment; first segment with one seta. Male antennule 18-segmented, geniculate between 15th and 16th segments and between 16th and 17th segments. Antenna consisting of coxa, basis, one-segmented small exopod, and three-segmented endopod; third endopodal segment terminating in large spine. Oral siphon extending over insertions of maxillipeds. Mandible consisting of thin stylet and indistinctly two-segmented palp tipped with two subequal setae. Maxillule bilobed; inner lobe about 3 times longer than outer lobe, tipped with four large setae; outer lobe tipped with three setae. Maxilla two-segmented, distal segment claw-like. Maxilliped consisting of syncoxa, basis, and four-segmented endopod. Leg 1 with three-segmented rami. Leg 2 with two-segmented rami. Leg 3 with obscurely segmented protopod, two-segmented exopod, lacking endopod. Leg 4 as vestigial lobe bearing two setae. Armature formula for legs 1–3 as follows: leg 5 represented by one seta on fifth pedigerous somite and one-segmented exopod bearing three setae.

	Coxa	Basis	Exopod	Endopod	
Leg 1	0-0	1-1	I-0; I-0; II, 2, 2	0-1; 0-1; 1, 2, 3	
Leg 2	0-0	1-0	I-0; I, II, 3	0-1; 1, 2, 2	
Leg 3	0-0	0-0	I-0; 0, II, 0	Absent	

Type species. Amalomyzon elongatum n. gen. n. sp. (original designation).

Etymology. The name of the new genus, Amalomyzon, as a new genus is a combination of Greek amal (= soft) and myz (=to suck), referring to the soft body of the type species. Gender neuter.

Remarks. Lee & Kim (2017) have listed 12 genera in the family Asterocheridae in which leg 4 is uniramous or represented by a lobe or a seta as in Amalomyzon n. gen. These genera comprise three genera (Coralliomyzon Humes and Stock, 1991, Cholomyzon Stock and Humes, 1969, and Temanus Humes, 1997). They were originally included in the family Coralliomyzontidae Humes and Stock, 1991. Most of these genera have a biramous leg 3. Only two genera, Andapontius Lee & Kim, 2017 and Holobinus Lee & Kim, 2017, have a uniramous leg 3 like Amalomyzon n. gen.

Amalomyzon n. gen. differs from genera Andapontius and Holobinus in having a 20-segmented female antennule with an aesthetasc on the antepenultimate segment (vs. 18-segmented female antennule with an aesthetasc on the terminal segment in both Andapontius and Holobinus). The new genus is further distinguished from Andapontius in having two-segmented endopod of leg 2 (vs. three-segmented in Andapontius) and two-segmented exopod of leg 3 (vs. 1-segmented in Andapontius). It can be distinguished from Holobinus in having a two-segmented exopod of leg 2 (vs. three-segmented exopod of leg 2 in Holobinus) and two spines plus four setae on the third exopodal segment of leg 4 (vs. two spines plus two setae in Holobinus). It should be noted that exopods of legs 3 and 4 of Andapontius are one-segmented. However, Lee & Kim (2017) have erroneously written two-segmented exopods of legs 3 and 4 of Andapontius. These and other differences between the new genus and its two related genera are summarized in Table 2.

Table 2 Differentiation of three asterocherid genera.

		Genus		
Characters	Andapontius Lee & Kim, 2017	Holobinus Lee & Kim, 2017	Amalomyzon n. gen.	
Body form	Broad	Narrow	Cylindridal	
Female antennule	18-segmented	18-segmented	20-segmented	
Armature of 3rd exopodal segment of leg 1	2 spines + 4 setae	2 spines + 2 setae	2 spines + 4 setae	
Leg 2 exopod	3-segmented	3-segmented	2-segmented	
Leg 2 endopod	3-segmented	2-segmented	2-segmented	
Leg 3 exopod	1-segmented	2-segmented	2-segmented	
Leg 4	1-segmented	As a lobe	As a lobe	

Amalomyzon elongatum n. sp.	
urn:lsid:zoobank.org:act:3CAFD973-9823-4672-8F19-14665C4D5CFD	
Figs. 2, 3 and 4	

Type material. Holotype (intact ♀; NIBR NIBRIV0000901203), intact paratypes (9 ♀ ♀, 2 ♂ ♂; NIBR NIBRIV0000901204), and dissected paratypes (1 ♀, 1 ♂) from washings of the sponge Dysidea dokdoensis Kang, Lee & Sim, 2020, SCUBA diving, at “Elephant Rock”, Ulleung Island, Korean East Sea (37°32′25″N, 130°50′51″E), a depth of 35 m, 29 Aug. 2018, coll. Jong Kuk Kim. Intact type specimens have been deposited in the National Institute of Biological Resources (NIBR), Incheon, Korea. Dissected paratypes are kept in the collection of I-H Kim.

Description

Female. Body (Figs. 2A and 2B) elongated, nearly cylindrical, indistinctly segmented. Body lengths of figured specimens 842 µm. Prosome 545 µm long, more than twice as long as wide, consisting of cephalosome and 4 indistinctly defined metasomal somites. Cephalosome circular, 227 × 264 µm, wider than metasomal somites. Metasome gradually narrowing posteriorly; metasomal somites defined by lateral constrictions between them. Urosome (Fig. 2C) 4-segmented, clearly defined from prosome. Fifth pedigerous somite 132 µm wide. Genital double-somite slightly wider than long (120 × 129 µm), widest at 30% region of double-somite length, narrowing posteriorly; genital apertures positioned dorsolaterally at 38% region of double-somite length just posterior to widest region. Abdomen incompletely articulated from genital double-somite. Two free abdominal somites (50 × 70 µm and 45 × 58 µm, respectively), both somites unornamented. Caudal ramus (Fig. 2D) 1.53 times longer than wide (29 × 19 µm), armed with 3 distal and 3 subdistal naked setae; 3 subdistal setae positioned at subequal planes; all setae longer than ramus.

Figure 2 Amalomyzon elongatum n. gen. n. sp., female.

(A) Habitus, dorsal. (B) Habitus, right. (C) Urosome, dorsal. (D) Right caudal ramus, dorsal. (E) Cephalic region, ventral. (F) Antennule. (G) Antenna. (H) Mandible. Scale bar: 0.1 mm in A and B, 0.05 mm in C and E, and 0.02 mm in D and F–H.

Figure 3 Amalomyzon elongatum n. gen. n. sp., female.

(A) Maxillule. (B) Maxilla. (C) Maxilliped. (D) Leg 1. (E) Leg 2. (F) Leg 3. (G) Leg 4 pair. (H) Leg 5. (I) Left genital aperture. Scale bar: 0.02 mm.

Figure 4 Amalomyzon elongatum n. gen. n. sp., male.

(A) Habitus, dorsal. (B) Urosome, ventral. (C) Antennule. (D) Leg 5 pair. Scale bar: 0.1 mm in A, 0.05 mm in B, and 0.02 mm in C and D.

Rostrum (Fig. 2E) small, spatulate, with convex distal margin. Antennule (Fig. 2F) 153 µm long, 20-segmented, gradually narrowing distally; armature formula 1, 2, 2, 2, 2/2, 2, 2, 5, 2/2, 2, 2, 2, 2/2, 2, 1+aesthetasc, 3, and 9; all setae naked, slender. Antenna (Fig. 2G) consisting of coxa, basis, exopod, and endopod; coxa short and unarmed; basis broadening distally, unarmed; exopod small, unsegmented, twice longer than wide (8 × 4 µm), armed with two distal and one middle setae; endopod three-segmented; first endopodal segment 31 µm long, unarmed but ornamented with spinules along outer margin; second endopodal segment small, with 1 seta; third endopodal segment 12 µm long, with two small setae subdistally, terminating in nearly straight spine (32 µm long).

Oral siphon (Fig. 2E) 193 × 43 µm, extending beyond insertions of maxilliped, not reaching insertions of legs 1. Mandible (Fig. 2H) consisting of stylet and palp; stylet 173 µm long, thin, its distal teeth hardly visible under microscope; palp 25 × 8 µm, much shorter than stylet, indistinctly two-segmented at distal quarter, tipped with two naked (setae 68 µm and 58 µm long, respectively). Maxillule (Fig. 3A) bilobed; inner lobe (precoxal endite) 64 × 20 µm, tapering distally, tipped with 4 long setae (longest seta: 120 µm), and ornamented with one row of spinules and one row of setules; outer lobe (palp), 20 × 5 µm, about one-third as long as inner lobe, with two distal setae (longer seta: 50 µm) and one subdistal seta. Maxilla (Fig. 3B) two-segmented; proximal segment unarmed; distal segment claw-like distal part, ornamented with one transverse row of fine setules at two-fifths region of segment length. Maxilliped (Fig. 3C) consisting of syncoxa, basis, and four-segmented endopod; syncoxa with one small seta at inner distal corner; basis being longest segment, unarmed, with parallel outer and inner margins; endopodal segments armed with zero, one, one, and one seta, respectively; terminal claw nearly straight, 25 µm long, 1.4 times longer than third endopodal segment.

Leg 1 (Fig. 3D) biramous, with three-segmented rami. Leg 2 (Fig. 3E) with 2-segmented rami. Leg 3 (Fig. 3F) with two-segmented exopod; endopod absent. Leg 4 (Fig. 3G) as small lobe bearing 1 short, spiniform apical seta and one longer, naked lateral seta. Inner coxal seta absent in legs 1–3. Inner distal corner of leg 1 produced. Outer spine on first exopodal segment large, 25 µm long, extending over midway of third exopodal segment. Articulation between second and third endopodal segments indistinct. Protopod of leg 3 indistinctly segmented. Exopodal segment of leg 4 about twice longer than wide (17 × 8 µm), armed with 3 naked setae distally. Armature formula for legs 1–3 as follows:

	Coxa	Basis	Exopod	Endopod	
Leg 1	0-0	1-1	I-0; I-0; II, 2, 2	0-1; 0-1; 1, 2, 3	
Leg 2	0-0	1-0	I-0; I, II, 3	0-1; 1, 2, 2	
Leg 3	0-0	0-0	I-0; 0, II, 0	Lacking	

Leg 5 (Fig. 3H) consisting of one lateral naked seta on fifth pedigerous somite and one-segmented exopod; exopodal segment 17 × 8 µm, with nearly parallel inner and outer margins, armed with three naked setae. Leg 6 (Fig. 3I) represented by one seta and one denticle on genital operculum.

Male. Body (Fig. 4A) as in female. Body length of figured specimen 580 µm long. Urosome (Fig. 4B) 5-segmented. Genital somite wider than long (80 × 110 µm), with convex lateral margins. Abdomen gradually narrowing posteriorly. Three free abdominal somites (48 × 66 µm, 36 × 59 µm, and 30 × 45 µm, respectively). Caudal ramus 1.67 times longer than wide (25 × 15µm). Rostrum as in female.

Antennule (Fig. 4C) 18-segmented, with geniculation between antepenultimate and penultimate segments; armature formula 1, 2, 2, 2, 2/2, 2, 1, 5, 2/1, 2, ?, ?, ?/?, 2+aesthetasc, and 8+aesthetasc (setations of thirteenth to sixteenth segments obscure due to overlaps of setae and segments). Antenna and mouthparts as in female.

Legs 1–3 as in female. Leg 4 (Fig. 4D) incompletely two-segmented; proximal segment with one seta laterally; distal segment unarmed or armed with one apical seta. Leg 5 (Fig. 3B) similar to that of female. Leg 6 (Fig. 4B) represented by two equal, small setae on distal apex of genital operculum.

Etymology. The specific name refers to the elongate body of the new species.

Remarks. The body form of Amalomyzon elongatum n. sp. is similar to that of Tuphacheres micropus Stock, 1965 known in the Mediterranean Sea (Stock, 1965). However, these two species differ from each other at the generic level as they exhibit different ways of leg segmentation reductions.

Dokdocheres n. gen.	
urn:lsid:zoobank.org:act:6787FECB-E7FA-48DB-97CF-73EBDEB4E151	

Diagnosis. Body broad, dorsoventrally flattened. Prosome consisting of cephalothorax and three metasomal somites. Urosome four-segmented, with two-segmented abdomen. Caudal ramus with six setae. Antennule 18-segmented, with aesthetasc on antepenultimate segment; first segment with one seta. Antenna consisting of coxa, basis, one-segmented exopod, and 2-segmented endopod; exopodal segment elongate, more than half length of first endopodal segment, with three setae; second endopodal segment terminating in slender claw. Oral cone short, stout. Mandible consisting of stylet bearing six teeth distally and palp tipped with one seta. Maxillule bilobed; inner lobe tipped with three large setae; outer lobe with three large and one small setae. Maxilla 2-segmented; distal segment as claw. Maxilliped consisting of syncoxa, basis, and 3-segmented endopod. Legs 1–4 biramous, with 3-segmented rami. Armature formula for legs 1–4 as follows:

	Coxa	Basis	Exopod	Endopod	
Leg 1	0-1	1-0	I-1; I-1; I+1, 2, 2	0-1; 0-2; 1, 1+I, 3	
Leg 2	0-1	1-0	I-1; I-1; II, I, 4	0-1; 0-2; 1, II, I+2	
Leg 3	0-1	1-0	I-1; I-1; II, I, 4	0-1; 0-2; 1, I, I+2	
Leg 4	0-0	1-0	I-1; I-1; II, I, 4	0-1; 0-2; 0, I, 2	

Leg 5 consisting of protopod bearing one seta and one-segmented exopod bearing 3 setae.

Type species. Dokdocheres rotundus n. gen. n. sp. (original designation).

Etymology. The name of the new genus is derived from Dokdo Island, the type locality of the type species, and -cheres, the ending of Asterocheres, the type genus of the family Asterocheridae. Gender masculine.

Remarks. The establishment of Dokdocheres n. gen. is justifiable by multiple extraordinary features in the leg setation of its type species, D. rotundus n. gen. n. sp., as follows: (1) the third exopodal segment of leg 1 is armed with 1 spine plus five setae (armature formula I+1, 2, 2), which is a unique armature condition in the family Asterocheridae; (2) the armature formula of the third endopodal segment of leg 1 is 1, 1+I, 3 (having 1 distal spine), which is a feature shared by three genera (Gomumucheres Humes, 1996, Parasterocheres Humes, 1996, and Phyllocheres Humes, 1996) within the family, all of which are associated with the sponge Dysidea in the Moluccas (Humes, 1996); (3) the armature formula of the third exopodal segment of legs 2–4 is II, I, 4. The same armature formula of the third exopodal segment is exhibited on leg 2 in 15 genera, leg 3 in 10 genera, and leg 4 in four genera. However, only Dokdocheres n. gen. and two existing genera (Siphonopontius Malt, 1991 and Stenomyzon Kim, 2010) exhibit the same armature condition in all legs 2–4; (4) the armature formula of the third endopodal segment of leg 4 is 0, I, 2. This armature condition is shared by five existing genera (Asterocheroides Malt, 1991, Cecidomyzon Stock, 1981, Cephalocheres Kim, 2010, Hammatimyzon Stock, 1981, and Scottomyzon Giesbrecht, 1897). Therefore, Dokdocheres n. gen. can be recognized by the above first feature alone or by the combination of the other three unusual features.

Dokdocheres rotundus n. sp.	
urn:lsid:zoobank.org:act:AB9D5F75-F663-4593-9058-5C5CA9F9B02A	
Figs. 5, 6 and 7	

Type material. Holotype (intact ♀; NIBR NIBRIV0000901205) and paratype (♀, dissected and mounted on a slide) from washings of the sponge Acanthella vulgata Thiele, 1898, SCUBA diving, Dokdo Island, Korean East Sea (37°14′44″N, 131°51′54″E), a depth of 22 m, 23 Apr. 2015, coll. Hyun Soo Rho. Holotype has been deposited in the National Institute of Biological Resources (NIBR), Incheon, Korea. Dissected paratype is kept in the collection of I.-H. Kim.

Description

Female. Body (Fig. 5A) broad, dorsoventrally flattened. Body length of dissected and figured paratype, 0.62 mm. Prosome subcircular, 518 × 443 µm, only slightly longer than wide. Cephalothorax 280 µm long. Third and fourth pedigerous somites with concave posterodorsal margin. All prosomal somites with round posterolateral corners. Urosome (Fig. 5B) small, four-segmented, occupying 16% of body length. Fifth pedigerous somite 62 µm wide, broadening posteriorly. Genital double-somite laterally expanded, rhomboidal, wider than long (58 × 89 µm), widest at 58% of double-somite length; genital apertures large, located dorsolaterally posterior to widest region. Abdomen two-segmented. First abdominal somite very short, 5 × 45 µm. Anal somite (second abdominal somite) 13 × 44 µm. Caudal ramus (Fig. 5C) 1.07 times longer than wide (16 × 15 µm), armed with 6 setae, ornamented with fine spinules on ventrodistal margin; setae III–VI pinnate, setae II and VII naked.

Figure 5 Dokdocheres rotundus n. gen. n. sp., female.

(A) Habitus, dorsal. (B) Urosome, dorsal. (C) Right side of abdomen, ventral. (D) Rostrum. (E) Antennule. (F) Antenna. (G) Oral cone, right. (H) Mandibular stylet. (I) Mandibular palp. Scale bar: 0.1 mm in A, 0.02 mm in B, E, F, H, and I, 0.01 mm in C, and 0.05 mm in D and G.

Figure 6 Dokdocheres rotundus n. gen. n. sp., female.

(A) Maxillule. (B) Maxilla. (C) Maxilliped. (D) Leg 1. (E) Leg 2. Scale bar: 0.02 mm.

Figure 7 Dokdocheres rotundus n. gen. n. sp., female.

(A) Endopod of leg 3. (B) Leg 4. (C) Leg 5. Scale bar: 0.02 mm.

Rostrum (Fig. 5D) longer than wide (90 × 74µm), widest at proximal quarter, tapering distally, with truncate distal apex. Antennule (Fig. 5E) gradually narrowing distally, 18-segmented; first segment longest, third segment second longest; armature formula 1, 2, 6, 2, 2/2, 6, 2, 2, 2/ 2, 2, 2, 2, 2/2+aesthetasc, 2, and 11; setae naked, relatively long, highly flexible. Antenna (Fig. 5F) consisting of coxa, basis, 1-segmented exopod, and 2-segmented endopod; coxa short, unarmed; basis with thin spinules on inner margin; exopod 4.50 times longer than wide (36 × 8µm), gradually broadening distally, about 68% as long as first endopodal segment, armed distally with 3 setae, 2 of them longer than exopodal segment; first endopodal segment 53 × 25 µm, unarmed, with setules on distal part of outer margin; second endopodal segment small, 16 × 9 µm, armed with 2 long subdistal setae, 2 min distal setae, terminating in slender claw of 53 µm long.

Oral cone (Fig. 5G) short, stout, with tuft of setules at distal apex of labrum and labium. Mandible consisting of stylet (Fig. 5H) and palp (Fig. 5I); stylet curved along distal third, with 6 teeth distally; palp small, unsegmented, 17 × 5 µm, tipped with 1 long seta; no articulation present between palp and distal seta. Maxillule (Fig. 6A) bilobed; inner lobe (precoxal endite) about 36 × 14 µm, tipped with three large, pinnate setae and ornamented with several spinules subdistally; outer lobe (palp) 41 × 10 µm, slightly longer than inner lobe, tipped with three large and one small setae. Maxilla (Fig. 6B) 2-segmented; proximal segment (syncoxa) broad, unarmed; distal segment (basis) as slender, strongly curved claw, unarmed, with rows of minute spinules at distal region. Maxilliped (Fig. 6C) 5-segmented, consisting of syncoxa, basis, and 3-segmented endopod; syncoxa unarmed; basis 88 µm long, with one long seta (41 µm long) at 48% region of inner margin; first to third endopodal segments with two, one and one setae, respectively; terminal segment 38 µm long, distal seta large, longer than segment; terminal claw 58 µm long, with 2 min spinules near tip.

Legs 1–4 (Figs. 6E, 7A and 7B) biramous, with 3-segmented rami. Inner coxal seta present in legs 1–3, absent in leg 4. Basis of leg 1 lacking inner distal element. Outer seta on basis of legs 1–4 large, naked. Second endopodal segment of legs 1–3 expanded. Spines on legs 1–4 well-developed, naked without serration. Distal processes on exopodal and endopodal segments well-developed, acutely pointed. Second endopodal segments with monocuspid outer distal process. Third exopodal segment of leg 1 with characteristic armature, with one spine plus five setae. Armature formula for legs 1–4 as follows:

	Coxa	Basis	Exopod	Endopod	
Leg 1	0-1	1-0	I-1; I-1; I+1, 2, 2	0-1; 0-2; 1, 1+I, 3	
Leg 2	0-1	1-0	I-1; I-1; II, I, 4	0-1; 0-2; 1, II, I+2	
Leg 3	0-1	1-0	I-1; I-1; II, I, 4	0-1; 0-2; 1, I, I+2	
Leg 4	0-0	1-0	I-1; I-1; II, I, 4	0-1; 0-2; 0, I, 2	

Leg 5 (Figs. 5B and 7C) two-segmented, consisting of protopod and one-segmented exopod; protopod, wider than long, not articulated from somite, with one long, naked seta (59 µm long) dorsodistally; exopodal segment 28 × 12 µm, with three weakly pinnate setae (45, 50, and 20 µm long, respectively, from dorsal to ventral). Leg 6 (Fig. 5B) represented by 1 seta and one spinule on genital operculum.

Male. Unknown.

Etymology. The name of the new species is derived from the Latin rotund (= round), alluding to the subcircular body form.

Genus Asterocheres Boeck, 1859	
Asterocheres banderaae n. sp.	
urn:lsid:zoobank.org:act:670BF905-E285-453E-96C0-56980A041CE7	
Figs. 8, 9 and 10	

Type material. Holotype (intact ♀; NIBR NIBRIV0000901206), intact paratypes (25 ♀♀, 2 ♂♂; NIBR NIBRIV0000901207), and dissected paratypes (2 ♀ ♀, 1 ♂) from washings of the sponge Petrosia corticata (Wilson, 1925), trimix SCUBA diving, a depth of 45.2 m, Dokdo Island, Korean East Sea, 06 Jul. 2022, coll. Taekjun Lee. Intact type specimens have been deposited in the National Institute of Biological Resources (NIBR), Incheon, Korea. Dissected paratypes are kept in the collection of I.-H. Kim.

Description

Female. Body (Fig. 8A) rather narrow. Body length 803 µm in dissected and figured specimen (756–810 µm in 10 measured specimens). Prosome 553 µm long, tapering posteriorly. Cephalothorax subcircular, wider than long (342 × 409µm). Second, third, and fourth pedigerous somites 309, 262, and 178 µm wide, respectively. All prosomal somites with rounded lateral corners. Urosome (Fig. 8B) 4-segmented. Fifth pedigerous somite 90 µm wide. Genital double-somite slightly wider than long (108 × 112µm), consisting expanded anterior half and tapering distal half; genital apertures large, positioned dorsally; lateral margin of posterior half (Fig. 8C) with row of about 10 (9–11) thin setules; dorsal and ventral surfaces lacking scales. Two free abdominal somites 45 × 53, and 38 × 48 µm, respectively. Caudal ramus slightly wider than long (21 × 22 µm), with 6 setae; 2 large mid-terminal setae (setae IV and V) stiff, almost not flexible; seta IV usually curved proximally and directed posterolaterally.

Figure 8 Asterocheres banderaae n. sp., female.

(A) Habitus, dorsal. (B) Urosome, dorsal. (C) Left side of genital double-somite. (D) Antennule. (E) Antenna. (F) Oral cone. (G) Mandible. (H) Maxillule. (I) Maxilla. (J) Maxilliped. Scale bar: 0.1 mm in A, 0.05 mm in B and F, 0.02 mm in C–E and G–J.

Figure 9 Asterocheres banderaae n. sp., female.

(A) Leg 1. (B) Inner distal seta on basis of leg 1. (C) Leg 2. (D) Leg 3. (E) Leg 4. (F) Leg 5. Scale bar: 0.02 mm.

Figure 10 Asterocheres banderaae n. sp., male.

(A) Habitus, dorsal. (B) Urosome, dorsal. (C) Antennule. (D) Maxilliped. (E) Endopod of leg 3. Scale bar: 0.1 mm in A, 0.05 mm in B, and 0.02 mm in C–E.

Rostrum absent. Antennule (Fig. 8D) 316 µm long, 20-segmented; armature formula 7 on ninth segment, 2+aesthetasc on antepenultimate segment, 11 on terminal segment, two on other segments. Antenna (Fig. 8E) consisting of coxa, basis, one-segmented exopod, and three-segmented endopod; basis 130 × 24 µm, ornamented with longitudinal row of minute spinules; Exopod small, 2.73 times longer than wide (12 × 4.4 µm), with two unequal distal and one middle setae; first endopodal segment 3.06 times longer than wide (55 × 18 µm) with fine setules along outer margin; second endopodal segment small with 1 stiff seta; third endopodal segment with 3 setae, terminating in spine of 42 µm long.

Oral cone (Fig. 8F) 150 × 68 µm, extending to insertions of maxilliped, with about 15 min denticles on distal regions of lateral margins. Mandible (Fig. 8G) consisting of slender stylet and palp; stylet 146 µm long; palp slender, 9.0 times longer than wide (45 × 5 µm), tipped with two setae, longer one 102 µm long, and shorter one 38 µm long; palp plus longer seta as long as stylet. Maxillule (Fig. 8H) bilobed; inner lobe 3.24 times longer than wide (55 × 17 µm), about three times longer than outer lobe, tipped with 5 setae (one of them minute), ornamented with long setules on inner surface, longest seta 76 µm long; outer lobe 2.25 times longer than wide (18 × 8 µm) tipped with three long and one short setae. Maxilla (Fig. 8I) stout, two-segmented; both segments unarmed; proximal segment proximally with flexible tube of maxillary gland; distal segment claw-like, as long as proximal segment, with row of fine spinules subdistally. Maxilliped (Fig. 8J) consisting of syncoxa, basis, and 4-segmented endopod; syncoxa with 1 seta subdistally on inner margin and row of minute spinules at outer distal corner; basis with 1 small, vestigial seta near middle of inner margin and minute spinules on outer margin; 4 endopodal segment with 2, 0, 1, and 1 seta, respectively; fourth endopodal segment 35 µm long, terminating in slightly curved claw of 58 µm long.

Legs 1–4 (Figs. 9A–9E) biramous, with 3-segmented rami. Inner coxal seta absent in legs 1 and 4, present in legs 2 and 3. Inner distal seta (Fig. 8B) on basis of leg 1 proximally inflated, 46 µm long, with perpendicular setules (or spinules) on proximal part of outer margin. Armature formula for legs 1–4 as follows:

	Coxa	Basis	Exopod	Endopod	
Leg 1	0-0	1-1	I-1; I-1; III, 2, 2	0-1; 0-2; 1, 2, 3	
Leg 2	0-1	1-0	I-1; I-1; III, I, 4	0-1; 0-2; 1, 2, 3	
Leg 3	0-1	1-0	I-1; I-1; III, I, 4	0-1; 0-2; 1, 1+I, 3	
Leg 4	0-0	1-0	I-1; I-1; III, I, 4	0-1; 0-2; 1, 1+I, 2	

Leg 5 (Fig. 9F) consisting of protopod bearing one dorsodistal seta and one-segmented exopod. Exopodal segment 1.82 times longer than wide (31 × 17 µm), armed with 3 setae and ornamented with spinules on dorsal surface; subdistal ventral seta naked, thin, 25 µm long; distal and subdistal dorsal setae 42 and 36 µm long, respectively, ornamented with stiff setules (or setule-like spinules). Leg 6 (Fig. 8B) represented by one small spinule and one naked seta on genital operculum.

Male. Body (Fig. 10A) similar in form to that of female. Body length 580 µm in dissected and figured specimen. Prosome 393 µm long. Cephalothorax 227 × 270 µm. Urosome (Fig. 10B) five-segmented. Fifth pedigerous somite 75 µm wide. Genital somite much wider than long (98 × 134 µm), with round lateral margins and concave posterodorsal margin, ornamented with triangular scales scattered on dorsal and lateral surfaces. Three abdominal somites 20 × 49, 18 × 49, and 27 × 47 µm, respectively. Caudal ramus 18 × 21 µm; seta IV directed posteriorly, unlike that of female.

Antennule (Fig. 10C) 17-segmented, geniculate between fifteenth and sixteenth segments; armature formula 2, 2, 2, 2, 2/2, 2, 2, 7, 2/2, 4, 2, 2, 2/1+aesthetasc, and 9; one distal seta on terminal segment spiniform, modified. Antenna as in female.

Oral cone, mandible, maxillule, and maxilla as in female. Maxilliped (Fig. 10D) segmented as in female, but inner subdistal element on syncoxa transformed to spine tipped on protrusion of syncoxa; second endopodal segment bearing one small seta.

Legs 1, 2, and 4 as in female. Leg 3 endopod (Fig. 10E) sexually dimorphic: second segment with 2 massive, spinulose spines (instead of setae in female); distal spine on third segment longer than that of female. Leg 5 as in female. Leg 6 (Fig. 10B) represented by two unequal setae on genital operculum.

Etymology. The new species is named in honor of Dr. Bandera for her contribution to the taxonomy of the genus Asterocheres.

Remarks. The genus Asterocheres is inconveniently large, comprising about 90 known species (WoRMS Editorial Board, 2022), many of which are incompletely described. Nevertheless, A. banderaae n. sp. is distinguishable from its congeners by its several characteristic morphological features. The coxa of leg 1 of the new species lacks an inner seta. According to Bandera & Conradi (2016), this feature is also present in three existing species of Asterocheres (A. eugenioi Bandera & Conradi, 2014, A. sarsi Bandera and Conradi, 2009, and A. trisetatus Kim, 2010). In A. eugenioi and A. sarsi, caudal rami are 1.5 or more times longer than wide with the oral cone extending to the insertion of leg 1 (Bandera & Conradi, 2014). In A. trisetatus, the third endopodal segment of leg 1 is elongated (Kim, 2010). These features of the three species are not applicable to A. banderaae n. sp.

The second endopodal segment of leg 3 is sexually dimorphic, armed with two pinnate setae in the female, but with two strong specialized spines in males. Sexual dimorphism in leg 3 has been reported in several species of Asterocheres, such as A. bahamensis Kim, 2010, A. nidorelliae Reyes-González & Suárez-Morales, 2021, A. peniculatus Kim, 2010, A. plumosus Kim, 2010, A. urabensis Kim, 2004, and A. walteri Kim, 2004 (Kim, 2004; Kim, 2010; Reyes-González & Suárez-Morales, 2021). However, in all these species, sexual dimorphism in leg 3 occurs on the third endopodal segment. None of them occurs on the second endopodal segment as in A. banderaae n. sp. They also differ from the new species in various other morphological aspects.

The syncoxa (first segment) of the male maxilliped of the new species bears a specialized inner distal spine. This feature is shared only with A. cuspis Kim, 2016 known in the eastern coast of Korea. Interestingly, like A. banderaae n. sp., A. cuspis has a posteriorly tapering body and a proximally swollen inner distal seta on the basis of leg 1. Despite these similarities, A. cuspis is distinguished from the new species by having a two-segmented mandibular palp (cf. 1-segmented in A. banderae n. sp.) and an inner coxal seta on leg 1 without sexual dimorphism on the second endopodal segment of leg 3.

Many species of Asterocheres were described based only on their females. Thus, the above characters (2) and (3) cannot be used for differentiating these species. However, A. banderaae n. sp. is readily recognizable by the above feature (1) and additional diagnostic features of females such as 1-segmented mandibular palp, similar lengths of the mandibular palp and its stylet, proximally swollen inner distal seta on the basis of leg 1, presence of about 10 setules on the lateral margin of the genital double-somite, and absence of an inner coxal seta on leg 4.

Figure 11 Scottocheres nesobius n. sp., female.

(A) Habitus, dorsal. (B) Urosome, dorsal. (C) Right caudal ramus, dorsal. (D) Antennule. (E) Antenna. (F) Proximal part of oral siphon. (G) Mandible. (H) Maxillule. (I) Maxilla. (J) Maxilliped. (K) Leg 1. (L) Leg 5. Scale bar: 0.1 mm in A, 0.05 mm in B, and 0.02 mm in C–L.

Figure 12 Scottocheres nesobius n. sp., female.

(A) Leg 2. (B) Leg 3. (C) Leg 4. (D) Left genital aperture. Male: (E) Habitus, dorsal. (F) Urosome, ventral. (G) Antennule. (H) Maxilliped. Scale bar: 0.02 mm in A-D, G, and H, 0.1 mm in E, and 0.05 mm in F.

Genus Scottocheres Giesbrecht, 1897	
Scottocheres nesobius n. sp.	
urn:lsid:zoobank.org:act:4ACBCDB8-4CA4-4803-ABA6-DAC84F4E7661	
Figs. 11 and 12	

Type material. Holotype (intact ♀; NIBR NIBRIV0000901208), intact paratypes (7 ♀ ♀; NIBR NIBRIV0000901209), and dissected paratypes (2 ♀ ♀, 1 ♂) from washings of the sponge Myxilla producta Hoshino, 1981, trimix SCUBA diving, a depth of 45.2 m, Dokdo Island, Korean East Sea, 06 Jul. 2022, coll. Taekjun Lee. Intact type specimens have been deposited in the National Institute of Biological Resources (NIBR), Incheon, Korea. Dissected paratypes are kept in the collection of I.-H. Kim.

Description

Female. Body (Fig. 11A) narrow. Body length 690 µm in dissected and figured specimen. Mean body length 702 µm (655–740 µm) based on seven specimens. Prosome 368 µm long. Cephalothorax 207 × 223 µm. Three metasomal somites (second, third, and fourth pedigerous somites) 189, 164, and 127 µm wide, respectively. Posterolateral corners of all prosomal somites rounded. Urosome (Fig. 11B) 4-segmented. Fifth pedigerous somite 95 µm wide. Genital double-somite 1.33 times longer than wide (118 × 89µm), with tooth-like process on lateral margin at anterior third; genital apertures positioned dorsally at anterior quarter. Two free abdominal somites 47 × 45 and 34 × 38 µm, respectively. Anal somite with two pairs of minute spinules on posteroventral margin near posteromedian incision (Fig. 11C). Caudal ramus (Fig. 11C) 6.07 times longer than wide (91 × 15 µm), widened at distal region, armed with 6 setae positioned distally and subdistally; setae IV-VI pinnate, other setae naked.

Rostrum absent. Antennule (Fig. 11D) 172 µm long, 17-segmented; armature formula 1, 2, 3, 1, 2/1, 2, 4, 2, 2/1, 2, 1, 1, 0/1+aesthetasc, and 12; setae small and naked; third segment with rudiment of partial articulation. Antenna (Fig. 11E) consisting of coxa, basis, one-segmented exopod, and two-segmented endopod; coxa short, unarmed; basis 50 µm long; exopod 8 × 4.5 µm, positioned distal quarter of endopod, armed with 3 thin setae (two unequal distal and one in middle); proximal endopodal segment 38 µm long, unarmed but ornamented with patch of minute spinules at inner subdistal region; distal endopodal segment with 4 small setae and terminating in slender, arched spine of 57 µm long.

Oral siphon (Fig. 11F) slender, extending to insertions of leg 4. Mandible (Fig. 11G) represented by thread-like stylet; palp absent. Maxillule (Fig. 11H) bilobed; inner lobe tapering distally, 28 × 13 µm, tipped with 2 long and 1 short setae (longest one 76 µm long) and ornamented with spinules on inner margin and distal region; outer lobe small, 13 × 2.5 µm, tipped with 2 unequal setae. Maxilla (Fig. 11I) slender, two-segmented; distal segment articulated at 60% region of segment length, with claw-like distal part. Maxilliped (Fig. 11J) slender, consisting of syncoxa, basis, and three-segmented endopod; syncoxa with one small seta at inner distal corner; basis 60 µm long, unarmed; first to third endopodal segments with two, one and one seta, respectively; third endopodal segment 27 µm long, terminating in long spine of 65 µm long.

Legs 1–4 (Figs. 11K, 12A–12C) biramous, with 3-segmented rami. Inner coxal seta absent in legs 1 and 2, but present in legs 3 and 4. Outer seta on basis small, naked. Inner distal spine on basis of first leg 21 µm long. Second endopodal segment of legs 1–3 with 1 inner seta, that of leg 4 with 2 inner setae. Armature formula for legs 1–4 as follows:

	Coxa	Basis	Exopod	Endopod	
Leg 1	0-0	1-I	I-1; I-1; II, 2, 2	0-1; 0-1; 1, 2, 3	
Leg 2	0-0	1-0	I-1; I-1; II, I, 4	0-1; 0-1; 1, 2, 3	
Leg 3	0-1	1-0	I-1; I-1; II, I, 4	0-1; 0-1; 1, I, 3	
Leg 4	0-1	1-0	I-1; I-1; II, I, 3	0-1; 0-2; 1, I, 2	

Leg 5 (Fig. 11L) 2-segmented; proximal segment (protopod) clearly articulated from somite, nearly triangular, 47 × 43 µm, with 2 setae (medial and lateral); distal segment (exopod) elliptical, 1. 75 times longer than wide (51 × 29 µm), with 3 small setae distally. Leg 6 (Fig. 12D) represented by 2 small setae and 1 spiniform process on genital operculum.

Male. Body (Fig. 12E) narrow. Body length of dissected and figured specimen 573 µm. Cephalothorax 180 × 182 µm. Urosome (Fig. 12F) 5-segmented. Genital somite 79 × 99 µm, nearly quadrangular, with notch on both sides of posterodorsal margin. Three abdominal somites 46 × 58, 36 × 45, and 27 × 37 µm, respectively, Caudal ramus 3.81 times longer than wide (61 × 16 µm).

Rostrum absent. Antennule (Fig. 12G) 182 µm long, 15-segmented, geniculate between antepenultimate and penultimate segments; armature formula 1, 2, 5, 2, 3/2, 2, 6, 3, 3/2, 3, 1, 3+aesthetasc, and 11; proximal seta on 12th segment minute, spinule-like. Antenna as in female.

Oral siphon, mandible, maxillule, and maxilla as in female. Maxilliped (Fig. 12H) slightly different from that of female in having angular protrusion at proximal quarter of inner margin of basis. Legs 1–5 as in female. Leg 6 represented by 3 small setae on genital operculum (Fig. 12F).

Etymology. The specific name nesobius is derived from Greeks neso (= island) and bio (= life), alluding to the discovery of the species in the island.

Remarks. The third exopodal segment of legs 2–4 of Scottocheres nesobius n. sp. bears two outer spines (armature formula II, I, 4 in legs 2 and 3, and II, I, 3 in leg 4). In the genus Scottocheres, these armature conditions are shared only with S. laubieri Stock, 1967 and S. mipoensis Kim, 2016. Otherwise, these two species distinctly differ from the new species as they have short caudal rami which are as long as wide or wider than long (Stock, 1967; Kim, 2016) with the third exopodal segment of leg 1 armed with three spines plus four setae (formula III, 2, 2) rather than two spines plus four setae (armature formula II, 2, 2) as in the new species.

Regarding the dimension of the caudal ramus, S. nesobius n. sp. appears to be similar to S. gracilis Hansen, 1923 as the latter species has elongated caudal rami which are five times longer than wide (Hansen, 1923). Scottocheres gracilis, a deep-sea species known from the southwest of Iceland, was obscurely described, with unknown leg armature. In S. gracilis, according to Hansen (1923), the caudal ramus is slightly shorter than the first and second free abdominal somites combined (“second and third abdominal somites combined” according to Hansen) (cf. distinctly longer in the new species). The body of the female is 1.05 mm long, which is significantly longer than that (about 0.7 mm long) of the new species. The exopodal segment of leg 5 extends beyond the tooth-like lateral process of the genital double-somite (cf. the exopodal segment terminates before the process in the new species).

Conclusions

Two new genera and four new species of copepods of the family Asterocheridae associated with sponges are described from the Korean East Sea (Sea of Japan). Hosts and differential characters of the new copepod species are as follows:

Amalomyzon elongatum n. gen. n. sp. is associated with Dysidea dokdoensis Kang, Lee & Sim, 2020. It bears an elongated body, two-segmented rami of legs 2, uniramous leg 3 with a two-segmented exopod, and rudimentary leg 4 represented by a lobe.

Dokdocheres rotundus n. gen. n. sp. is associated with Acanthella vulgata Thiele, 1898. It has an 18-segmented female antennule, 2-segmented endopod of the antenna, and unusual setations of swimming legs including the presence of three spines plus four setae on the third exopodal segment of legs 2–4.

Asterocheres banderaae n. sp. is associated with Petrosia corticate (Wilson, 1925). It bears no inner coxal seta on leg 1 or 4, but has two strong, sexually dimorphic inner spines on the second endopodal segment of male leg 3.

Scottocheres nesobius n. sp. is associated with Myxilla producta Hoshino, 1981. It is characterized by elongate female caudal rami about six times longer than wide, a 17-segmented female antennule, and an armature of two spines plus four setae on the third exopodal segment of leg 1.

After adding the two new genera and four new species described here, the family Asterocheridae in the Korean East Sea now comprises 12 species in seven genera.

We thank Dr. Jimin Lee (Korea Institute of Ocean Science and Technology) for providing copepod samples for us to study.

Additional Information and Declarations

Competing Interests

Author Contributions

Data Availability

New Species Registration

The authors declare there are no competing interests.

Il-Hoi Kim conceived and designed the experiments, performed the experiments, analyzed the data, prepared figures and/or tables, authored or reviewed drafts of the article, and approved the final draft.

Taekjun Lee conceived and designed the experiments, performed the experiments, authored or reviewed drafts of the article, and approved the final draft.

The following information was supplied regarding data availability:

The morphological characteristics, and taxonomical status of all new species and genera are available in the ‘Results & Discussion’.

The following information was supplied regarding the registration of a newly described species:

Publication LSID: urn:lsid:zoobank.org:pub:0E84F5CE-D4C5-4AAD-A72C-CCBD77806CDB

Amalomyzon: urn:lsid:zoobank.org:act:214F5279-8986-4F6A-A32E-FD3BCF0EA638

Amalomyzon elongatum: urn:lsid:zoobank.org:act:3CAFD973-9823-4672-8F19-14665C4D5CFD

Dokdocheres: urn:lsid:zoobank.org:act:6787FECB-E7FA-48DB-97CF-73EBDEB4E151

Dokdocheres rotundus: urn:lsid:zoobank.org:act:AB9D5F75-F663-4593-9058-5C5CA9F9B02A

Asterocheres banderaae: urn:lsid:zoobank.org:act:670BF905-E285-453E-96C0-56980A041CE7

Scottocheres nesobius: urn:lsid:zoobank.org:act:4ACBCDB8-4CA4-4803-ABA6-DAC84F4E7661.

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
