# Peer review of "Two new genera and four new species of Asterocheridae (Copepoda: Siphonostomatoida) associated with sponges (Porifera) from the Korean East Sea"

_PeerJ, doi:10.7717/peerj.14889_

## Round 0.1 · original submission · Major Revisions

The paper describes four new species, therefore, it is absolutely important because it contributes to the estimation of biodiversity. Furthermore, the type of microhabitat is very interesting and it increases the value of this paper.

Reviewer 1 ·

Basic reporting

Present manuscript describes Four new copepod species including two new genera. As alpha taxonomy, it has enough and important findings in copepod taxonomy. I found only few sleeps of pen on the manuscript. After revision, present paper should be published in PEERJ.

Experimental design

no specific issue. Present paper follows standard method for taxonomy.

Validity of the findings

No specific comments

Additional comments

Please check few errors as follows:
line 525 Bears --> bears
line 628 put "." after "ventral"
line 633 come --> cone.

·

Basic reporting

.

Experimental design

.

Validity of the findings

.

Additional comments

All copepodologists know well that Il-Hoi Kim’s drawings are excellent and here also provided
with full clarity and perfect for the manuscript on “Four new species of Asterocheridae (Copepoda,
Siphonostomatoida) associated with sponges (Porifera) from the eastern coast of Korea”.
I have comments mainly in writing.

Abstract: Authors have written the armature formula in the abstract, which need to be
deleted and replace with clear features which define the new genera and new species. Hence,
the current abstract needs to be totally revised by proving only important features, not the
armature formula. For each species, please provide atleast 3-4 characteristic features and
provide, differs from which genera or species. For ex. Authors have given as Asterocheres has
20-segmented antennule, well, Amalomyzon also has the same. It will be interesting for the
readers if you provide clear differences with other species. Provide information in the
abstract regarding copepod, siphonostomatoid, not only in the title. Even can write each
species associated with sponges, their names and systematics.

Introduction: In Korea, 20 species of the Asterocheridae are reported. Please provide this
information about 20 species in table format, not in writing format. Table will be helpful for
the readers. Even, for reviewers, it is difficult to understand. Please add the hosts
information, it is not provided anything in the introduction and their ecology bit.
Lines 66-68 in methods, need to be revised. How deep it was and where exactly collected, and
the methodology of collection by proving photos will be interesting for the readers. New
researchers can understand exactly how to collect the samples and also preparation and
preservation techniques. Here, I would like to see the map as Fig. 1 and scuba diving pictures
as Fig. 2. It will stimulate the readers to see the taxonomic paper in PeerJ journal.
Line 107-108 : I am not clear, please revise.
107 geniculate between segments 15, 16, 17fifteenth and sixteenth and sixteenth and
seventeenth segments
Line 132-133, cf. 18-segm.... Please provide the references. Instead of “cf.” better to revise
as “vs.”
Line 140-141: better to keep a table of comparison between Lee & Lim 2017 with this genus.
Type material collected from Ullung Island.. It’s better to provide the map for all species in 1
figure.
L 119: Leg 3 endopod – vestigial or absent?

Remarks:
Better to provide the information in table format. Can add some information about the
importance of hosts, it will be helpful for the readers to understand more about the copepods
and its hosts.
Conclusion, still more clarity is needed, may mention much about the hosts
All species descriptions have done well, however, need some changes as pointed out for the
abstract, introduction, remarks and conclusion.

Reviewer 3 ·

Basic reporting

This good taxonomic work needs to be published after a major revision. Need a thorough grammatical check, better with the help of a fluent English speaker. The description and illustrations are very good. But the Introduction and abstracts need a revision (please see my comments)

Experimental design

need minor revisions

Validity of the findings

Need to be published after a major revision

Annotated reviews are not available for download in order to protect the identity of reviewers who chose to remain anonymous.

---

## Round 0.2 · accepted · Accept

The authors addressed all the requests made by reviewers in the several steps of the revision so it is my opinion that the paper can be accepted in its present form.

Reviewer 1 ·

Basic reporting

It is now good enough to be published in peerJ, since authors follow comments throughly by my review suggestions.

Experimental design

no problems.

Validity of the findings

It has enough new findings.

Additional comments

no further comments.

·

Basic reporting

Manuscript has been revised well. Sufficient information has been provided in relation to the 2 new genera including 4 new species from Sea of Japan, Korea.

Experimental design

It is sufficient and map is provided, which gives more clarity to understand the field area.

Validity of the findings

It's very important for the science. Many species are undiscovered in marine area and new species findings are always interesting and useful for the science and young researchers.

Additional comments

I commend the authors for their excellent work.
In the abstract, it is mentioned as female antennule, also line 119, why so?
What is the difference between male and female antennule?
Only for 1 genus, male has been recorded, not for all species.

Korean East Sea is acceptable? or may I suggest - East Sea of Korea?

My suggested title is

Two new genera and four new species of copepods of the family Asterocheridae (Siphonostomatoida) associated with sponges (Porifera) from the East Sea of Korea.

Reviewer 3 ·

Basic reporting

Good and acceptable.
The authors thoroughly revised the MS by considering all the issues highlighted in the previous version.

Experimental design

Good and acceptable.

Validity of the findings

It should be published.

Additional comments

Now it is in acceptable form.